# Feedbacks between sea-floor spreading, trade winds and precipitation in the Southern Red Sea

Kurt Stüwe [1] ✉, Jörg Robl [2], Syed Ali Turab [3], Pietro Sternai[4] & Finlay M. Stuart[5]

Feedbacks between climatic and geological processes are highly controversial and testing them is a key challenge in Earth sciences. The Great Escarpment of the Arabian Red Sea margin has several features that make it a useful natural laboratory for studying the effect of surface processes on deep Earth. These include strong orographic rainfall, convex channel profiles versus concave swath profiles on the west side of the divide, morphological disequilibrium in fluvial channels, and systematic morphological changes from north to south that relate to depth changes of the central Red Sea. Here we show that these features are well interpreted with a cycle that initiated with the onset of spreading in the Red Sea and involves feedbacks between orographic precipitation, tectonic deformation, mid-ocean spreading and coastal magmatism. It appears that the feedback is enhanced by the moist easterly trade winds that initiated largely contemporaneously with sea floor spreading in the Red Sea.

Feedbacks between geological and surface processes are an intriguing discovery that often explain persistent issues in Earth sciences[1]. In particular the interaction between climatic and tectonic processes has been in sharp focus in past years[2–6]. Recent modelling studies have suggested that feedback can develop between orographic precipitation, mantle melting and rifting processes[7,8].

The Red Sea rift system initiated with the emplacement of the Afar plume underneath northeastern Africa at about 31 Ma[9]. While this probably caused initial updoming and erosion in the region, separation of the two plates did not occur until about 25 Ma[10]. Rifting is likely to have been asymmetric with more surface uplift and a thinner mantle lithosphere on the eastern (Arabian) side[11,12]. Nevertheless, on both sides of the Red Sea, rift flank uplift and consequential erosion caused denudation of the entire 1000 m thick Mesozoic sediment pile[13] that covered most of the Proterozoic basement in northern Africa and the Arabian peninsula before the onset of rifting (Fig. 1). A single Mesozoic sediment outcrop is

preserved at almost 3000 m elevation on top of Saudi Arabia's highest peak, Mt. Al Soudah (Fig. 1d). It places tight constraints on the total amount of rock uplift and erosion, at least for the escarpment edge region (Fig. 1d). Although the Arabian plate separated from Africa at 25 Ma[10], it was not until later (5 – 13 Ma[9,11,14]) that oceanic lithosphere formed in the central Red Sea rift (Fig. 1d). Prior to the formation of oceanic crust, the widening Red Sea was underlain by attenuated continental lithosphere of both adjacent plates. The entire rift history was accompanied by volcanic activity that occurred almost exclusively on the Arabian side (the Saudi Arabian Harrats[12]). This volcanic activity has generally increased in intensity in the last three million years[15,16]. East of the Red Sea, the Saudi Arabian rift flank forms one of the most spectacular escarpments in the world. The difference in χ either side of the divide[17] (Fig. 1b) increases to the south most likely indicating increasing erosion rate differences and a tendency of the divide to migrate east. The edge of the escarpment forms the continental divide for much of its length and

[1]Institut für Erdwissenschaften, Universität Graz, Graz, Austria. [2]University of Salzburg, Geography and Geology, Salzburg, Austria. [3]National Centre of Excellence in Geology, University of Peshawar, Peshawar, Khyber Pakhtunkhwa, Pakistan. [4]Department of Earth and Environmental Sciences, University of Milano-Bicocca, Milan, Italy. [5]Isotope Geoscience Unit, Scottish Universities Environmental Research Centre, East Kilbride, UK. ✉e-mail: kurt.stuewe@uni-graz.at

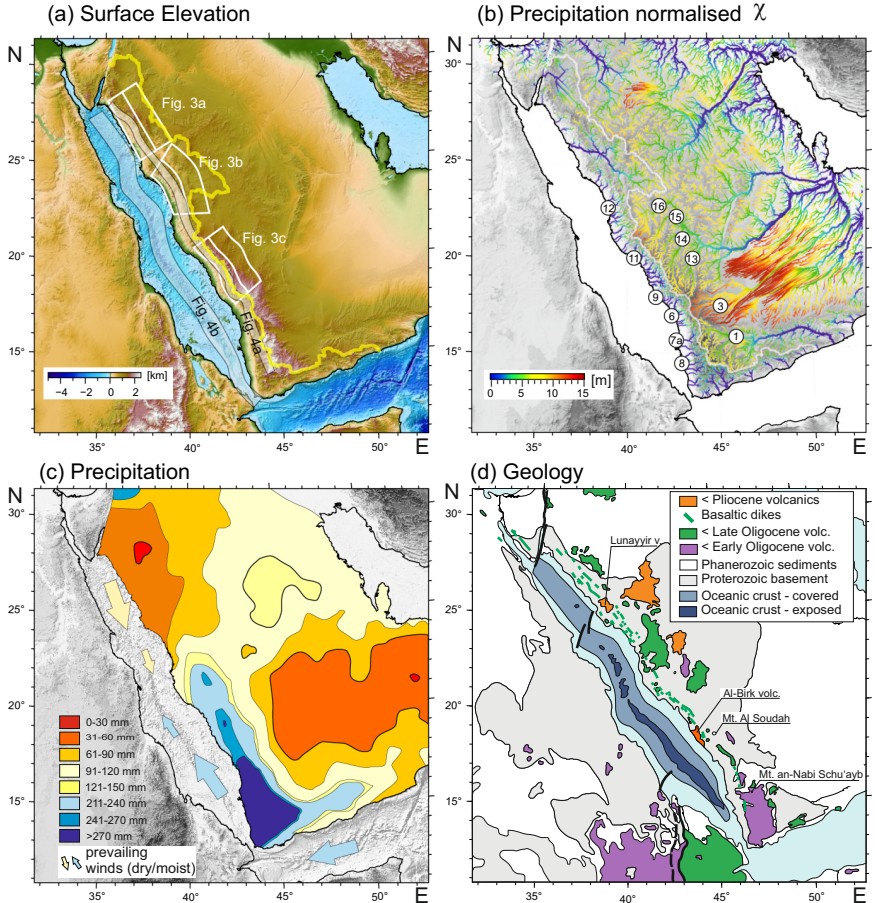

**Fig. 1 | Major features of the Saudi Arabian margin. a** Topography also showing the continental divide (yellow line) and the position of the curved swath profiles shown in Figs. 3 and 4 (white boxes and shaded region along Red Sea center). **b** χ[17] map normalized for precipitation (method[35], precipitation data[24], DEM data[36].

Numbered labels are wadis for which channel profiles are shown in Fig. 2. **c** Precipitation averaged over 50 year interval[24,25]. **d** Geology (distribution of volcanics after ref. 15, oceanic lithosphere after ref. 9).

separates a flat inland landscape from a kilometer high drop to the Red Sea. The escarpment elevation increases southward reaching highest point (3666 m) at Yemen's highest peak Jabal an-Nabi Shu'ayb. This southern part of the escarpment is characterized by strongly orographic rainfall pattern with the western, coastal side of the escarpment receiving almost 300 mm/year more than the high elevation eastern region (Fig. 1c).

Here we show that the climate, geomorphology and geological history of the Red Sea and the adjacent Arabian margin are intrinsically linked. We suggest that the feedback between climate and mantle melting is kept running, at least in part, by the easterly trade winds and may result in future acceleration of the rifting process.

## Results

Various models have been proposed to explain the evolution of the passive margin and the formation of the escarpment[18,19]. However, the margin shows a series of interesting spatial and temporal correlations that suggest an interdependency between deep and shallow processes that have not been recognised to date.

### Channel profiles

There is an intriguing difference between those draining east and those draining west. Variation in steepness index $k_{sn}$ (a measure to quantify the potential of a river to incise into its bedrock) can be used as a proxy for the degree of geomorphic equilibration, e.g., ref. 20 (Fig. 2). In general the degree of disequilibrium increases

southward. In the south, where erosion-driven flexural rebound would be expected in response to the escarpment erosion on the coastal side of the divide (from swath profiles and geochronological data, Fig. 3c), channel profiles are largely graded showing only a broad, low amplitude disequilibrium bulge in $k_{sn}$ (Figs. 1b, and 2 left column). In contrast, east draining channels show a high amplitude bulge in $k_{sn}$ some 50–100 km east of the drainage divide suggesting that they were affected by higher uplift rates in that region (Fig. 2). Curiously, the shape of the east draining channels is not reflected in the mean topography, which is concave in profile perpendicular to the escarpment (Fig. 3c). The nature of the channel disequilibria on both the east and west side of the divide, separated by 100 km, suggest separate formation phases.

**Mean topography.** Curved swath profiles[21] with baseline following the escarpment edge show a systematic topographic change from north to south along the Saudi Arabian margin (Fig. 3). In the northern and central part they show a broad convex mountain range (Fig. 3a, b). However, in the southern part, where topography and orographic rainfall is highest, mean topography is concave on both sides of the escarpment (Fig. 3c) so that the continental divide coincides with the escarpment edge at most places. This is typical of the shape of a divide that is caused by scarp retreat or scarp degradation and associated erosion-driven flexural rebound processes[22,23]. The concave shape of the mean topography (Fig. 3c) contrasts the convex bulge in the channel profiles in particular on the inland (east) side of the divide (Fig. 2).

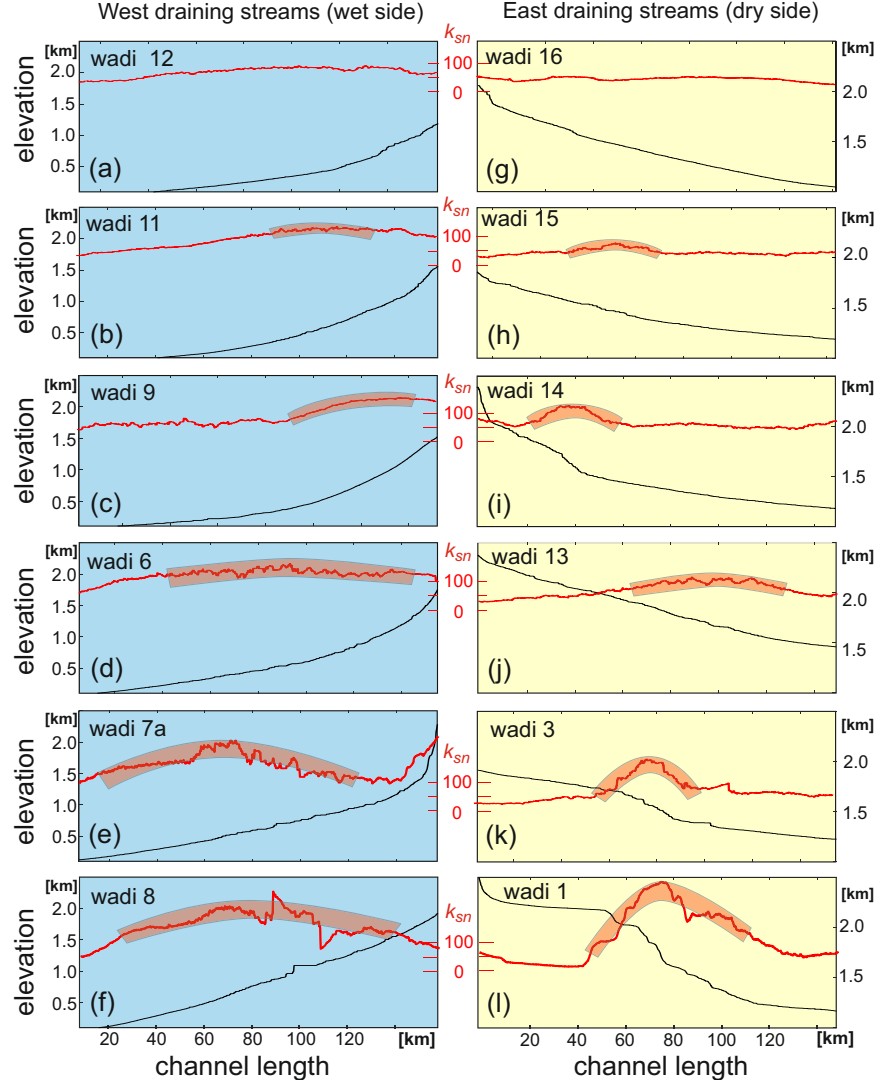

**Fig. 2 | Channel profiles of wadis along the Saudi Arabian Red Sea margin.** Blue for west draining streams (wet side), yellow for east draining streams (dry side). Note that due to the sinuosity of wadis, the distance along the channels does not necessarily corresponds to the distance from the coastal margin. Location of wadis is shown in Fig. 1b and is generally from north (at top) to south (at bottom). Red shaded regions of channel profiles show geomorphic disequilibrium sections of wadis. **a–f** presents geomorphic disequilibrium due to erosion driven Pliocene flexural rebound in wadi 12, 11, 9, 6, 7a and 8. **g–l** present geomorphic dis-equilibrium due to Miocene uplift in response to downbending margin in wadis 16, 15, 14, 13, 3 and 1.

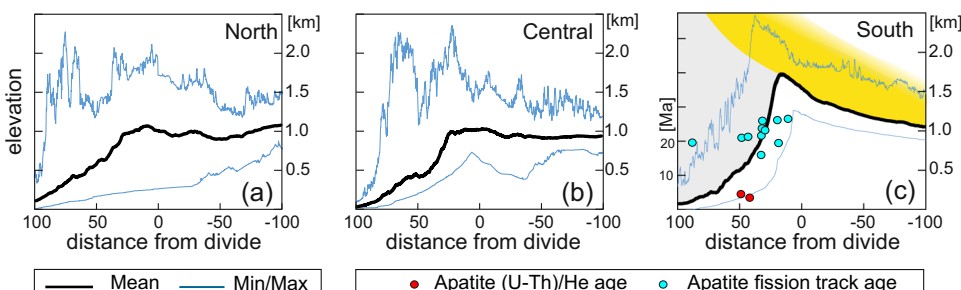

**Fig. 3 | Curved topographic swath profiles[21] across the Red Sea margin.** Swaths are for the northern (in a), central (in b) and southern part (in c) of the escarpment and are derived from the SRTM3_V4 data[36]. Locations of profiles are shown in Fig. 1a. The profile for the study region is shown in **c** and also includes the low temperature geochronological data of Turab et al. (in review). The shaded region in this panel shows the total amount of denudation (yellow = Phanerozoic sediments, grey = basement rocks). Black line is the mean, blue are minimum and maximum values.

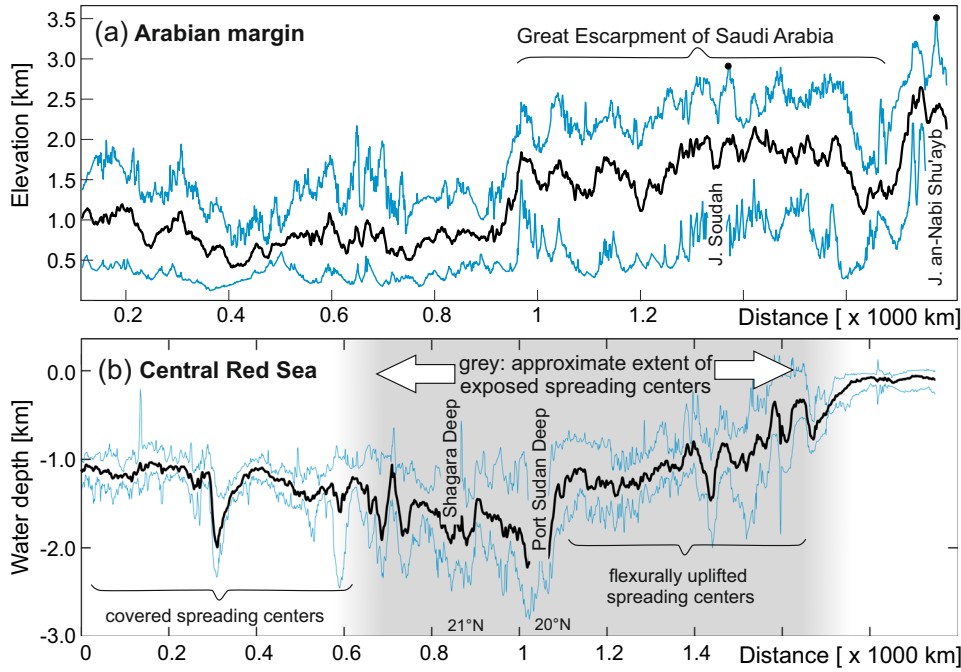

**Fig. 4 | Curved topographic swath profiles[21] along the Red Sea margin.** Swaths are near the divide along the Saudi Arabian margin (in a) and near the spreading centers in the Central Red Sea (in b) and are derived from the SRTM15plus data[37]. Locations of profiles are shown in Fig. 1a. Black line is the mean, blue are minimum and maximum values.

**Total denudation.** The total denudation since the Oligocene across the divide is spatially heterogeneous. The existence of isolated outcrops of Mesozoic sediments inland of the divide on both sides of the Red Sea indicates that the amount of denudation there is comparable to the former thickness of the sediments and has barely penetrated the basement (Figs. 1d and 3c). This sediment thickness has been estimated to be of the order of 500–1000 m in the region[13] and can be considered as the total amount of rift-related denudation inland of the divide. In contrast, in the region immediately west of the Arabian continental divide, denudation rises abruptly to at least four kilometers, as suggested by low temperature thermochronological data[19] and by simple geometric analysis of the topography (Fig. 3c).

**Topographic relationship to Red Sea depth.** Although several hundreds of kilometers apart, the elevation of the Saudi Arabian margin shows some relationship to the depth of the central Red Sea (Fig. 4). In particular, there is no significant escarpment developed in the region where spreading centers are covered and not observed and the continental divide is located significantly further inland (Fig. 1a). Conversely, the escarpment is well developed where spreading centers are observed. Going south along the escarpment towards Yemen, the spreading centers in the central Red Sea are located at successively shallower levels (Fig. 4a, b).

**Precipitation.** Although much of the Arabian peninsula is dry, less than 100 mm precipitation per year, the coastal region of southern Saudi Arabia and Yemen has a 50 year average precipitation rate that exceeds 270 mm per year[24,25] (Fig. 1c). The region of high precipitation rates correlates closely with the region where the escarpment is more than 2000 m high and is almost exclusively restricted to its western side of the escarpment where the prevailing easterly trade winds reach the coast. In contrast, much of the northern Red Sea coast is dominated by dry northerly winds[26]. A comparison with model predictions for the Pliocene (www.paleoclim.org) shows that the precipitation pattern was similar, with possibly an even stronger precipitation contrast across the divide[27]. The modern wind and precipitation pattern emerged at about 13 Ma (e.g., ref. 28) and hence

contemporaneously with the onset of oceanic lithosphere formation in the center of the Red Sea.

**Low temperature thermochronology.** As the total amount of denudation during and post-rifting was only a few kilometers, the main low-temperature thermochronometric systems do not provide clear timing of erosional events[19]. Nevertheless, some rocks from elevations around 200 m below the highest regions of the escarpment were exhumed through the apatite fission track partial annealing zone (120–70 °C) at 15 Ma and 25 Ma[19]. They clearly record erosion during rift flank uplift. Two apatite (U-Th)/He ages (He) (recording cooling through about 80–50 °C) from the same locations record ages of 2 Ma and 3 Ma (Fig. 3c). The large age gap is difficult to explain by slow cooling and exhumation. Based on thermal modelling Turab et al. (in review) suggest a two phase exhumation history separated by 10–15 Myr when little erosion occurred. They suggest that the He ages record cooling due to an exhumation phase in the Pliocene, which was characterized by escarpment retreat. This interpretation is supported by low temperature thermochronology from the Eritrean margin where a similar, albeit shorter, age gap records scarp retreat in the younger part of the erosion history[18].

## Discussion

In the southern Red Sea the surface topography, the cooling history of the upper crust and the decompression melting of the mantle may be explained by a process in which rifting is strongly coupled with orographic precipitation.

We suggest that before the initial formation of oceanic lithosphere in the center of the Red Sea, the topography of the margins evolved as typical rift flank margins. Broad zones of surface uplift on both sides of the Red Sea formed slightly asymmetric mountain ranges with the Arabian margin being slightly higher, due to thinner mantle lithosphere on that side[11]. Erosion of the uplifted rift flanks initially caused removal of about one kilometer of the Mesozoic sediments on both sides of the rift (Fig. 5a)[18,19]. This first phase of erosion substantially slowed when the sediments were removed and the erosionally resistant Proterozoic basement rocks reached the surface.

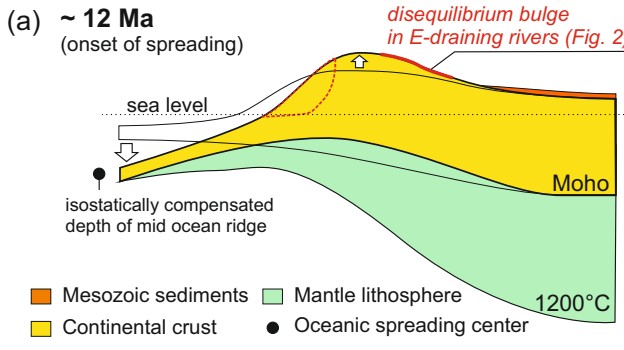

(a) **~ 12 Ma**
(onset of spreading)

*disequilibrium bulge in E-draining rivers (Fig. 2)*

sea level

isostatically compensated depth of mid ocean ridge

Moho

1200°C

■ Mesozoic sediments    ■ Mantle lithosphere
■ Continental crust       ● Oceanic spreading center

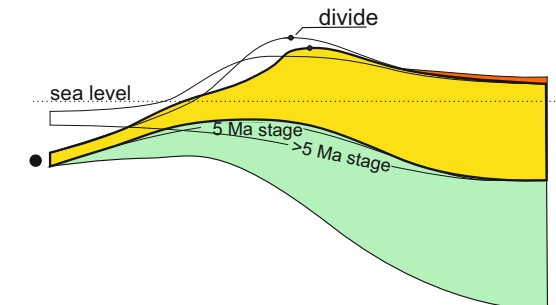

(b) **Today: Without orographic rainfall**

divide

sea level

5 Ma stage
>5 Ma stage

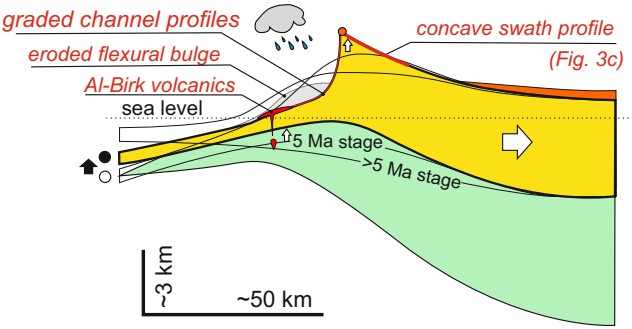

(c) **Today: With orographic rainfall**

*graded channel profiles*
*eroded flexural bulge*
*Al-Birk volcanics*
sea level

*concave swath profile*
*(Fig. 3c)*

5 Ma stage
>5 Ma stage

~3 km    ~50 km

**Fig. 5 | Major steps in the evolution of the Arabian Red Sea margin. a** At or near the time of initial formation of oceanic lithosphere. Formation of oceanic lithosphere in Red Sea downwarps margin and creates additional uplift by flexural doming (future erosion dashed red). **b, c** Evolution since the initial formation of oceanic lithosphere. **b** Without orographic rain fall. A minor escarpment forms and erosion of uplifted area causes downwearing and backwards mirgrating escarpment with minor flexural rebound in foreland, escarpment not exactly at divide (not realized or only realized in northern, dry part of Red Sea depending on age of spreading there). **c** With orographic rainfall as for the southern Red Sea coast. Erosion causes additional uplift at escarpment edge, escarpment at divide, decompressional melting in foreland where flexural bulge is eroded softens lithosphere. Flexural lifting of the spreading center occurs.

Initial formation of oceanic lithosphere (12–13 Ma[14]) caused the Red Sea to deepen to some 2000 m, being the isostatically compensated depth of mid-ocean ridges (Fig. 5a). This caused downbending of the continental margin to this depth and consequential additional flexural updoming of uplifted margin as described by Turcotte et al.[29] (Fig. 5a). This updomed bulge bent channel profiles draining inland (Fig. 2, right column) and also caused final dessication of the climate east of the divide allowing channels to retain their profile. On the west side of the divide, the newly created topography caused increased orographic precipitation resulting in rapid erosion of a crescent-

shaped region (red dashed outline on Fig. 5a) in the Pliocene. This second phase of erosion involved the formation of the pronounced escarpment. This caused flexural rebound resulting in further uplift near and below the escarpment forming a concave mean topography (Fig. 3c) on the east side[23]. On the west side, erosion largely compensated the rebound bulge (grey shaded crescent on Fig. 5c) with only a broad $k_{sn}$ bulge being visible in the channel profiles (Fig. 2 left column). This second phase of rapid erosion also resulted in the deposition of substantial sediment sequences that are known to be much younger than the rifting age[30].

The evolution proposed above reflects a strong coupling between tectonic and climatic processes. We propose that there is a positive feedback between: (i) The formation of new oceanic lithosphere in the Red Sea causing renewed rift flank uplift and orographic precipitation and (ii) orographic precipitation causing asymmetric denudation of the updomed region, flexural rebound and ultimately accelerated spreading. Clearly, the former is easily explained: the close spatial correlation of precipitation with the high topography of the Arabian rift flank caused by the rifting in connection with the prevailing moist trade winds coming from the Indian Ocean makes it plausible that feedback between increasing orographic precipitation and ridgeline uplift arises.

However, the precipitation has also affected the mechanics of spreading. We propose two mechanisms allowed the localised rainfall at rift flanks to influence the formation of mid-ocean spreading in the central Red Sea: Firstly, precipitation and consequential Pliocene erosion causes substantial flexural doming of the escarpment edge and rock uplift in the foreland. This erosion-driven rock uplift also uplifts the uppermost mantle lithosphere causing decompression melting of the younger Harrats, for example the <1 Ma Al-Birk volcanics[15] (Figs. 1d and 5c). These volcanics have been suggested to have formed at about 70 km depth in equilibrium with Harzburgite inclusions and thus at about 1200 °C[31] and it is known that there is a decrease in melting depth of the volcanics from east to west[32]. It is therefore plausible that the westernmost <1 Ma old coastal volcanism is a result of decompression at shallow mantle levels in response to flexural rebound, rather than in relation to the Afar plume. Such volcanism may cause thermal weakening of the lithosphere thus aiding the spreading. Secondly, the flexural rebound of the escarpment caused by erosion created a mountain range of some 3000 m surface elevation (Fig. 4a). This caused flexural uplift of the mid-ocean spreading centers (e.g., south of 20°N, Fig. 4b). The higher elevation of the spreading centers in this region may have assisted further spreading via their increased potential energy.

We test the feedback model suggested above with a thermomechanical model that couples a description of surface processes like erosion with the mechanics of lithospheric extension[7,8]. We explore the mechanical consequences of melting caused by orographic precipitation in by tracking the lithospheric extension rate. Specifically, we use an extension rate of 2 cm per year a potential mentle temperature of 1200 °C and a crustal thickness of 35 km (Fig. 6 of Sternai[7]) and extract mechanical model results from this simulation for time steps that pertain to the Red Sea rifting. In particular, we explore how the extensional strain rate changes in response to increased surface erosion, rock uplift and mantle melting. Figure 6a shows the mantle melt volume created by decompression melting in response to extension for two surface erosion rates characterized by the effective erosional diffusivity $k$. The higher erosion rates (red curve) cause substantially more decompression melting in the mantle than lower erosion rate. The consequence of these processes to lithospheric extension are shown in Fig. 6b, c where the vertically averaged second invariant of the strain rate tensor is plotted as a proxy for mean rifting rate. Approximately 3 Myr after rifting in the model (we relate rifting in the model to the onset of oceanic lithosphere formation in the Red Sea at 13 Ma so that 3 Myr in the model run corresponds to about 10 Ma in

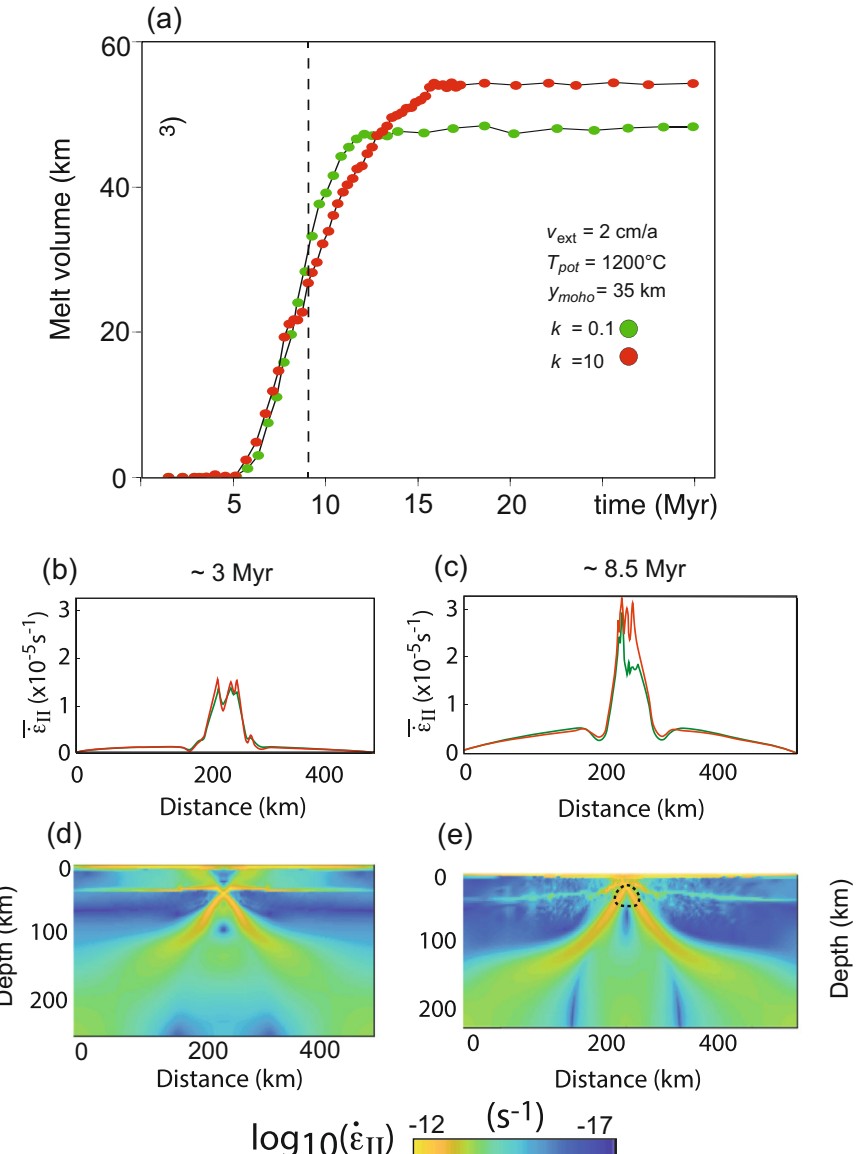

**Fig. 6 | Thermomechanically coupled numerical simulation of continental extension with parameters adjusted to reflect the Red Sea opening.** The results are for an extension rate of $v_{ext} = 2$ cm per year, a potential mantle temperature $T_{pot} = 1200\,°C$ and a Moho depth of $y_{moho} = 35$ km as well as two different effective erosional diffusivities $k$. Shown time steps correspond to million years after rifting and thus correspond in the Red Sea region to about 10 Ma and the Pliocene after onset of oceanic lithosphere formation at 13 Ma. **a** Mantle melt volume created by the extension. **b**–**e** Strain rate at two different time steps. **b**, **c** Vertically averaged second invariant of the strain rate tensor as a proxy for extension rate and stress decrease by melting. Note that the strain rate increases after 8 Myr when melting has become significant and that strain rates are higher for larger contribution of surface processes ($k = 10$ vs. $k = 0.1$). **d**, **e** colour coded strain rate.

the Red Sea), there is comparatively little difference in rifting rate for low or high surface erosion (red and green curves on Fig. 6b), but that the vertical redistribution of melt causes significant differences in extensional strain rate after 8.5 Myr (i.e., for the Red Sea in the Pliocene).

This model explains many of the north-south changes in morphology including escarpment profile (Fig. 3a–c) and the inland departure of the drainage divide from the escarpment lip north of about 23°N, where also the spreading centers vanish (Fig. 1a, d). Our model is consistent with a model that invokes a northwards propagation of the Red Sea spreading center[33,34]. In contrast, Augustin et al.[14] have shown that the Red Sea may have spread continuously along its length with most of the northern part being obscured by sediments (salt). This conflict can be explained by reflecting upon the distribution of orographic precipitation in the Red Sea region. The southern part of the Red Sea is characterized by moist trade

winds, while north of 20°N the wind changes by 180° and is dominated by dry northwesterlies. The increased escarpment elevation in the south may be related to the direction and moisture of the trade winds in the southern part of the Red Sea confining the processes described here to this region[19] (Fig. 1c). In this context it is interesting to note that a series of recent studies have documented sudden onset of the Indian Monsoon and thus the trade winds around 13 Ma (e.g., ref. 28), which is roughly contemporaneously with the proposed onset of formation of oceanic lithosphere in the central Red Sea.

We propose that the strong connection between orographic precipitation at the southern Red Sea coast, escarpment elevation, and mid-ocean spreading, is in line with morphological peculiarities and the rock cooling history. This requires feedback between the rainfall-caused erosion on the rifting process via (a) isostatically-driven uplift of the mantle lithosphere causing decompression melting and

softening of the crust; (b) flexural uplifting of the escarpment edge and the spreading center in the Red Sea, thus adding potential energy to the rift environment relative to central Saudi Arabia which ultimately affected rifting rate. In this context, the post-12 Ma basaltic volcanism in western Saudi Arabia need not be explained by the Afar plume, but by decompression melting in the mantle lithosphere caused by erosion-driven flexural uplift.

## Methods

The methods used for this article involved the plotting of topographic metrics using data and codes listed below and the interpretation of previously published data as listed in the text. Our thermomechanical modelling is based on the studies of Sternai[7] and Sternai et al.[8] who showed conceptually that orographic precipitation and lithospheric melting during continental extension may be coupled, using both analytical and numerical analyses. Here we use their two-dimensional cross-sectional thermomechanical model coupled with petrological considerations to track mantle melting in response to extension and erosion at the surface. The rheology of the model lithosphere is a visco-elasto-plastic rheology based on rock mechanical data and is modelled with a power law stress and exponential temperature dependence. Melting is modelled with a simple model considering solidus and liquidus temperatures. Surface processes are parameterized through the effective erosional diffusivity solving the equations of mass and energy conservation.

## Data availability

The results presented here are all derived from freely available digital elevation models that are for download from https://srtm.csi.cgiar. org/ and data presented herein. Climate data for Fig. 1c were downloaded from https://www.worldclim.org/data/worldclim21.html.

## Code availability

Topographic metrics were derived using the generic mapping tool GMT (https://www.generic-mapping-tools.org) as well as with codes written by the authors. These codes are available from the authors. The thermomechanical code used for the mechanical modelling is also available from the authors.

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

## Acknowledgements

We thank the Saudi Geological Survey SGS for field support, E. Sizova and S. Schorn for discussions of the Arabian igneous petrology, T. Wagner for help with data management and R. Burger and P. Kucera for discussion of the climate data. P.S. was supported by the project Dipartimenti di Eccellenza 2018-2022 of the Department of Earth and Environmental Sciences at the University of Milano-Bicocca funded by MIUR. The authors acknowledge the financial support by the University of Graz.

## Author contributions

K.S. contributed idea and writing of the manuscript, J.R. contributed the numerical modelling of the geomorphology figures, S.A.T. contributed field work, drafting maps and constructing swath profiles, P.S. contributed the thermomechanical modelling, F.S. contributed the interpretation of the geochronology.

## Competing interests

The authors declare no competing interests.
