## [Peer Review File · Nature Communications]

REVIEWER COMMENTS

Reviewer #1 (Remarks to the Author):

In the ms. "A Feedback Cycle Between Sea-Floor Spreading, Trade Wind and Rainfall" by Stuwe et al. as it is expressed by the title, the authors put in relation high precipitation rate along the southern Arabian margin, inducing erosion and related flexural rebound causing decompressional melting and emplacement of younger Harrats. And such volcanism causes thermal weakening of the lithosphere aiding the spreading. To support their idea, authors describe some topographic, morphological, geochronological characteristics of the southern Arabian margin that can be explained also advocating other more surficial processes, like considering different styles of topographic evolution of a high elevated rifted margin (see points below).

I personally found fascinating the idea that there is a feedback mechanism between high precipitation rate, erosion, and the mechanics of spreading. The idea that surface processes can have a strong influence on magmatism and the rifting processes is on fashion these days. But I really believe that some support has to be found in the numerical modeling models (see for instance the paper by P. Sternai, in Scientific Reports, 2020) or at least try with some flexural models. Otherwise, it is too qualitative and remains just a nice story.

1. paragraph called total denudation:

The authors stated that due to the presence of remnants of Mesozoic sediments inland of the water divide there is a heterogeneous distribution of the amounts of denudation with higher amounts seaward of the water divide. What they describe is typical for an escarpment evolving following a scarp retreat or plateau downwearing style, while it does not happen when it evolves following the downwarped style that the authors choose without discussion. The escarpment retreat and the plateau degradation evolutionary styles involve the presence of normal faults producing the scarp. This is certainly true for the Saudi Arabia margin to the north of the study area and it has been supposed for the facing Eritrean margin. I am not saying that it must be the same in the study area because it is not that the same evolutionary style has to be foreseen for a whole margin but at least it has to be discussed, even more, because the other two styles, respect to the one the authors choose, better explain some characteristics of the margin illustrated by the authors.

2) Paragraph mean Topography

The reported concave shape topography for the southern margin (Fig. 3c) is typical for the margin evolving following a plateau degradation model (see fig. 11 c of Brown, Gallagher, Johnson 1998, Annual Rev. Earth Planet. Sci 26, 519-572. This last evolutionary style implies a certain amount of erosion on the plateau inland of the drainage divide, and this could even explain in part the high amplitude bulge of the channels flowing east.

2) Paragraph Geochronology

The presence of thermochronology data with age close or younger than rifting is typical of either scarp retreat than plateau degradation model while they are unusual for the downwarping style (see again fig.11 c and related text in Brown, Gallagher, Johnson 1998, Annual Rev. Earth Planet. Sci 26, 519-572). Anyway, a set of samples transversal to the margin are needed to have a better idea about the style of the margin evolution, downwarping vs. scarp retreat or plateau downwarping, and, maybe, to discriminate between the last two that have a lot of common characters.

Moreover, authors stated that: "Thermal modelling of this data suggests slow denudation rates in the Miocene and rapid cooling by escarpment retreat in the Pliocene". First, I have to say that there is no access to these data, nor even to the channel analysis data. I asked for the supplementary material and there are not. Reviewers and readers in this way have no possibility to judge the soundness of the data-set. And there is no possibility to know how many (U-Th_Sm)/He data have been performed. In fig. 3c there are two points (2 samples?). Is it possible that they were reset in Pliocene times because of some other thermal events? (see for example young samples in Ghebreab W, Carter A, Hurford AJ, Talbot CJ. 2002. Earth and Planetary Sciences Letters 200: 107– 119). Have AFT analysis been applied on these 2 samples, too? In this way, it is difficult to get an idea.

Moreover, in Szymanski et al. 2016 Tectonics, 35, 2863-2895, for the Central Arabian margin, on the basis of apatite (U-Th_Sm)/He, they hypothesized a second extensional pulse that is older (about 16 Ma) with respect to what was obtained by the authors (ca. 2 Ma) but maybe what (U-Th_Sm)/He analyses are indicating is, in this case, too, another extensional pulse. Again, I am not saying that it is in this way but just that authors have to discuss this.

Another possibility is that, as the 2 points are placed 50 km inland (figure 3c) the reappraisal of the denudation is related to a change of style from plateau degradation to scarp retreat, due to the fact that one style evolves in the other, and in the scarp retreat there is a much-focussed erosion.

In paragraph 4 you stated: "Erosion of the uplifted rift flanks caused the removal of about one kilometer of the Mesozoic sediments on both sides of the rift (Fig. 4b) resulting in closure or partial closure of the apatite fission track systems in exposed rocks". But to expose totally reset samples for the AFT you need, even with a very high geothermal gradient (unusual, inland of a rifted margin, of 40°C/km at least two kms of removed overburden.

Reviewer #2 (Remarks to the Author):

“A Feedback Cycle Between Sea-Floor Spreading, Trade Wind and Rainfall

” by Stuwe et al., is a very example of a simple geological model constrained with a diverse dataset. The topographic analysis is clean and concise, providing enough information to follow the results. Incorporating the arguments from thermochronology and removal of sedimentary cover is crucial for the interpretation and logical. The overall results have important implications for future work and our understanding of tectonics and climate coupling. I would suggest minor revisions.

In the summary there is the clause: “large age differences between different low-T thermochronometers”. I was left not knowing what this means. Are there large age differences at the same location between thermochronometers? This means that the exhumation rate is very slow. Does it mean that the thermochron ages on either side of the rift are very different? It is just not clear to me.

End of introduction: “in part the caused by the trade winds”. This is a typo.

Start of Geological background: “long been of interest” this is the second time this phrase is used. Consider something else.

“postdate to rifting” this should just be postdate rifting.

Just before fluvial channels: “that show a peculiar spatial or temporal correlations ” should be “that show peculiar spatial...”

At the end of the Geochron section: “Thermal modelling of this data suggests slow denudation rates in the Miocene and rapid cooling by escarpment retreat in the Pliocene.” It is not clear what this is referring to. If there are geochron data that are important for the story, they should be included in the study not simply referred to. There is no way for the reader to assess this. In addition, in the Acknowledgements there is the sentence “SUERC and F. Stuart are thanked for assistance(sic) with the geochronological data that will form the focus on a subsequent paper by the third author.” It seems strange that F. Stuart would not be included in this study as a co-author if there is work that contributed to the story here. Of course, this is entirely up to the authors.

The model is clear and concise and I don't think it needs more editing or information. The figures are all clear as well.

Justification of reviewers comment

(Justifications are in red below the comments of the reviewers (in black))

REVIEWER COMMENTS

Reviewer #1 (Remarks to the Author):

In the ms. "A Feedback Cycle Between Sea-Floor Spreading, Trade Wind and Rainfall" by Stuwe et al. as it is expressed by the title, the authors put in relation high precipitation rate along the southern Arabian margin, inducing erosion and related flexural rebound causing decompressional melting and emplacement of younger Harrats. And such volcanism causes thermal weakening of the lithosphere aiding the spreading. To support their idea, authors describe some topographic, morphological, geochronological characteristics of the southern Arabian margin that can be explained also advocating other more surficial processes, like considering different styles of topographic evolution of a high elevated rifted margin (see points below).

* Yes, this is what its about ;-)

I personally found fascinating the idea that there is a feedback mechanism between high precipitation rate, erosion, and the mechanics of spreading. The idea that surface processes can have a strong influence on magmatism and the rifting processes is on fashion these days. But I really believe that some support has to be found in the numerical modeling models (see for instance the paper by P. Sternai, in Scientific Reports, 2020) or at least try with some flexural models. Otherwise, it is too qualitative and remains just a nice story.

This is the most critical point of the reviewer and was accommodated by quantifying our model for the Red Sea margin with a thermomechanical model of Sternai. A new section is inserted into the Manuscript.

1. paragraph called total denudation:

The authors stated that due to the presence of remnants of Mesozoic sediments inland of the water divide there is a heterogeneous distribution of the amounts of denudation with higher amounts seaward of the water divide. What they describe is typical for an escarpment evolving following a scarp retreat or plateau downwearing style, while it does not happen when it evolves following the downwarped style that the authors choose without discussion. The escarpment retreat and the plateau degradation evolutionary styles involve the presence of normal faults producing the scarp. This is certainly true for the Saudi Arabia margin to the north of the study area and it has been supposed for the facing Eritrean margin. I am not saying that it must be the same in the study area because it is not that the same evolutionary style has to be foreseen for a whole margin but at least it has to be discussed, even more, because the other two styles, respect to the one the authors choose, better explain some characteristics of the margin illustrated by the authors.

Of course we are aware of the 3 morphology shapes and we are actually convinced that the escarpment in the study region evolves by scarp retreat or plateau degradation as in other parts of the Red Sea. The confusion may have arisen due to our misuse of the word "downwarp". In the terminology of Gallagher et al (1998) or Balestrieri (2005) the word "downwarp" is a morphological

term

2) Paragraph mean Topography

The reported concave shape topography for the southern margin (Fig. 3c) is typical for the margin evolving following a plateau degradation model (see fig. 11 c of Brown, Gallagher, Johnson 1998, Annual Rev. Earth Planet. Sci 26, 519-572). This last evolutionary style implies a certain amount of erosion on the plateau inland of the drainage divide, and this could even explain in part the high amplitude bulge of the channels flowing east.

This is accommodated by our careful accommodation of the point above

2) Paragraph Geochronology

The presence of thermochronology data with age close or younger than rifting is typical of either scarp retreat than plateau degradation model while they are unusual for the downwarping style (see again fig.11 c and related text in Brown, Gallagher, Johnson 1998, Annual Rev. Earth Planet. Sci 26, 519-572). Anyway, a set of samples transversal to the margin are needed to have a better idea about the style of the margin evolution, downwarping vs. scarp retreat or plateau downwarping, and, maybe, to discriminate between the last two that have a lot of common characters.

Moreover, authors stated that: "Thermal modelling of this data suggests slow denudation rates in the Miocene and rapid cooling by escarpment retreat in the Pliocene". First, I have to say that there is no access to these data, nor even to the channel analysis data. I asked for the supplementary material and there are not. Reviewers and readers in this way have no possibility to judge the soundness of the data-set. And there is no possibility to know how many (U-Th_Sm)/He data have been performed. In fig. 3c there are two points (2 samples?). Is it possible that they were reset in Pliocene times because of some other thermal events? (see for example young samples in Ghebreab W, Carter A, Hurford AJ, Talbot CJ. 2002. Earth and Planetary Sciences Letters 200: 107– 119). Have AFT analysis been applied on these 2 samples, too? In this way, it is difficult to get an idea.

Moreover, in Szymanski et al. 2016 Tectonics, 35, 2863-2895, for the Central Arabian margin, on the basis of apatite (U-Th_Sm)/He, they hypothesized a second extensional pulse that is older (about 16 Ma) with respect to what was obtained by the authors (ca. 2 Ma) but maybe what (U-Th_Sm)/He analyses are indicating is, in this case, too, another extensional pulse. Again, I am not saying that it is in this way but just that authors have to discuss this.

Another possibility is that, as the 2 points are placed 50 km inland (figure 3c) the reappraisal of the denudation is related to a change of style from plateau degradation to scarp retreat, due to the fact that one style evolves in the other, and in the scarp retreat there is a much-focussed erosion.

We indicated in the acknowledgements of the first version that we were in the process of writing a paper on the geochronology. This manuscript is now in review with EPSL. The manuscript does not discuss any feedback processes as in the manuscript here, but derives a two phase erosion history of the escarpment that is used herein. In view of this other manuscript we substantially reduced the presentation and discussion of geochron data here. Conversely, we now discuss more and other previously published data from the region (that subsumed the paper by Ghebreab et al., suggested by the reviewer).

In paragraph 4 you stated: "Erosion of the uplifted rift flanks caused the removal of about one kilometer of the Mesozoic sediments on both sides of the rift (Fig. 4b) resulting in closure or partial closure of the apatite fission track systems in exposed rocks". But to expose totally reset samples for the AFT you need, even with a very high geothermal gradient (unusual, inland of a rifted

margin, of 40°C/km at least two kms of removed overburden.

Reviewer #2 (Remarks to the Author):

- All comments by reviewer 2 below were minor and were subsumed within our corrections of the comments of reviewer 1 above

“A Feedback Cycle Between Sea-Floor Spreading, Trade Wind and Rainfall” by Stuwe et al., is a very example of a simple geological model constrained with a diverse dataset. The topographic analysis is clean and concise, providing enough information to follow the results. Incorporating the arguments from thermochronology and removal of sedimentary cover is crucial for the interpretation and logical. The overall results have important implications for future work and our understanding of tectonics and climate coupling. I would suggest minor revisions.

In the summary there is the clause: “large age differences between different low-T thermochronometers”. I was left not knowing what this means. Are there large age differences at the same location between thermochronometers? This means that the exhumation rate is very slow. Does it mean that the thermochron ages on either side of the rift are very different? It is just not clear to me.

End of introduction: “in part the caused by the trade winds”. This is a typo.

Start of Geological background: “long been of interest” this is the second time this phrase is used. Consider something else.

“postdate to rifting” this should just be postdate rifting.

Just before fluvial channels: “that show a peculiar spatial or temporal correlations” should be “that show peculiar spatial...”

At the end of the Geochron section: “Thermal modelling of this data suggests slow denudation rates in the Miocene and rapid cooling by escarpment retreat in the Pliocene.” It is not clear what this is referring to. If there are geochron data that are important for the story, they should be included in the study not simply referred to. There is no way for the reader to assess this. In addition, in the Acknowledgements there is the sentence “SUERC and F. Stuart are thanked for assistance(sic) with the geochronological data that will form the focus on a subsequent paper by the third author.” It seems strange that F. Stuart would not be included in this study as a co-author if there is work that contributed to the story here. Of course, this is entirely up to the authors.

The model is clear and concise and I don't think it needs more editing or information. The figures are all clear as well.

REVIEWER COMMENTS

Reviewer #1 (Remarks to the Author):

I am glad that the authors insert a new section containing an extract of the Sternai model on extensional setting, and is clever the link they made with the reappraisal in the last million years of the Younger Harrat.

The authors postpone most of the denudation affecting the margin after the inception of seafloor spreading, the latter following recent publications is likely placed at 13 Ma. I do not completely agree with this because for example in Yemen, on the Red Sea side, AFT ages between 16 and 21 Ma are showing a long mean track length and thus are considered almost totally reset. To obtain these data, it is necessary that even in the first rifting phase (the real one, because after the inception of seafloor spreading is no more rifting) a considerable amount of cover has to be already removed.

My suggestion to overcome this problem is, for sure, to delete Fig. 4a representing the margin before 12 Ma, and this is because of multiple reasons: (1) there is represented only the uplift of the margin and not the erosion on the western side of the flank, toward the sea, that in part had to be ongoing due to the AFT ages (2) the authors trace the remnant of the Jurassic cover all through the margin (even on the western side) indicating a long wavelength flexure of the lithosphere and a margin topography characterized by a broad monocline; this implies that they choose a downwarp model for the formation of the margin (downwarp now it is not in a geomorphic sense but as an evolutionary model) (3) they do not need this figure because the shape in black and white is present in fig. 4b and it is more general without the necessity of too much not verified assertions.

This time the language is less accurate. Maybe the ms. would deserve another general look by the English native authors of the ms. and the single sections need to be revised by the pertaining authors. I have noted some minor incorrectness through the text in a pdf file.

For the new section 4.2

I recall the attention of the authors and in particular of P. Sternai on this sentence: "who showed conceptually that orographic precipitation and lithospheric melting during continental extension may be coupled, using both analytical and numerical analyses. Here, we expand these studies to explore the mechanical consequences of melting caused by orographic precipitation to the lithospheric extension rate".

I understand what you mean but in this way, jumping the intermediate processes it seems to me too excessive.

The last perplexity is that the model starts at the onset of rifting and the comparison with the Red Sea is with the onset of seafloor spreading.....

Justification of Reviewers comments

(authors justification in red)

All minor comments (spelling mistakes and the like) on the pdf annotated by the reviewer are now corrected

REVIEWER COMMENTS

Reviewer #1 (Remarks to the Author):

I am glad that the authors insert a new section containing an extract of the Sternai model on extensional setting, and is clever the link they made with the reappraisal in the last million years of the Younger Harrat.

Thank you

The authors postpone most of the denudation affecting the margin after the inception of seafloor spreading, the latter following recent publications is likely placed at 13 Ma. I do not completely agree with this because for example in Yemen, on the Red Sea side, AFT ages between 16 and 21 Ma are showing a long mean track length and thus are considered almost totally reset. To obtain these data, it is necessary that even in the first rifting phase (the real one, because after the inception of seafloor spreading is no more rifting) a considerable amount of cover has to be already removed.

My suggestion to overcome this problem is, for sure, to delete Fig. 4a representing the margin before 12 Ma, and this is because of multiple reasons: (1) there is represented only the uplift of the margin and not the erosion on the western side of the flank, toward the sea, that in part had to be ongoing due to the AFT ages (2) the authors trace the remnant of the Jurassic cover all through the margin (even on the western side) indicating a long wavelength flexure of the lithosphere and a margin topography characterized by a broad monocline; this implies that they choose a downwarp model for the formation of the margin (downwarp now it is not in a geomorphic sense but as an evolutionary model) (3) they do not need this figure because the shape in black and white is present in fig. 4b and it is more general without the necessity of too much not verified assertions.

OR, you are probably right (although this could be discussed at length) but we thank you for suggesting a straightforward way to overcome this by deleting Fig. 4a – this is now done.

This time the language is less accurate. Maybe the ms. would deserve another general look by the English native authors of the ms. and the single sections need to be revised by the pertaining authors. I have noted some minor incorrectness through the text in a pdf file.

Done

For the new section 4.2

I recall the attention of the authors and in particular of P. Sternai on this sentence: “who showed conceptually that orographic precipitation and lithospheric melting during continental extension may be coupled, using both analytical and numerical analyses. Here, we expand these studies to explore the mechanical consequences of melting caused by orographic precipitation to the lithospheric

extension rate". I understand what you mean but in this way, jumping the intermediate processes it seems to me too excessive.

Fair enough – we expanded and reworded this section.

The last perplexity is that the model starts at the onset of rifting and the comparison with the Red Sea is with the onset of seafloor spreading.....

Thanks for noting this. This was indeed an error in our wording and is now corrected

7.6.2022 Kurt Stuewe